# The Phylogeny and Metabolic Potentials of a Lignocellulosic Material-Degrading *Aliiglaciecola* Bacterium Isolated from Intertidal Seawater in East China Sea

**DOI:** 10.3390/microorganisms12010144

**Published:** 2024-01-11

**Authors:** Hongcai Zhang, Zekai Wang, Xi Yu, Junwei Cao, Tianqiang Bao, Jie Liu, Chengwen Sun, Jiahua Wang, Jiasong Fang

**Affiliations:** 1Shanghai Engineering Center of Hadal Science and Technology, College of Marine Sciences, Shanghai Ocean University, Shanghai 201306, China; zhanghongcai2021@163.com (H.Z.); m220200648@st.shou.edu.cn (Z.W.); xyu@shou.edu.cn (X.Y.); jwcao@shou.edu.cn (J.C.); d230200051@st.shou.edu.cn (T.B.); d220200055@st.shou.edu.cn (J.L.); sunchengwen2022@163.com (C.S.); 2Laboratory for Marine Biology and Biotechnology, Qingdao National Laboratory for Marine Science and Technology, Qingdao 266237, China

**Keywords:** marine microorganisms, *Aliiglaciecola*, whole genome, lignocellulosic materials, biodegradation

## Abstract

Lignocellulosic materials are composed of cellulose, hemicellulose and lignin and are one of the most abundant biopolymers in marine environments. The extent of the involvement of marine microorganisms in lignin degradation and their contribution to the oceanic carbon cycle remains elusive. In this study, a novel lignin-degrading bacterial strain, LCG003, was isolated from intertidal seawater in Lu Chao Harbor, East China Sea. Phylogenetically, strain LCG003 was affiliated with the genus *Aliiglaciecola* within the family *Alteromonadaceae*. Metabolically, strain LCG003 contains various extracellular (signal-fused) glycoside hydrolase genes and carbohydrate transporter genes and can grow with various carbohydrates as the sole carbon source, including glucose, fructose, sucrose, rhamnose, maltose, stachyose and cellulose. Moreover, strain LCG003 contains many genes of amino acid and oligopeptide transporters and extracellular peptidases and can grow with peptone as the sole carbon and nitrogen source, indicating a proteolytic lifestyle. Notably, strain LCG003 contains a gene of dyp-type peroxidase and strain-specific genes involved in the degradation of 4-hydroxy-benzoate and vanillate. We further confirmed that it can decolorize aniline blue and grow with lignin as the sole carbon source. Our results indicate that the *Aliiglaciecola* species can depolymerize and mineralize lignocellulosic materials and potentially play an important role in the marine carbon cycle.

## 1. Introduction

Lignocellulosic material (LCM) is a class of complex biopolymers on Earth [1]. It is the most abundant plant organic matter, forming 50% of the plant biomass. LCM consists of three main components: cellulose, hemicellulose and lignin [2,3]. Lignin is the second-most abundant terrestrial organic material after cellulose, accounting for 30% of marine sedimentary organic matter [4]. Traditionally, LCM is considered as hard-to-biodegrade organic matter. Most of the hard-to-biodegrade organic compounds contain structures such as single or multiple fused benzene rings, heterocycles and long-chain alkanes and are characterized by high stability and low hydrophilicity [5,6]. Rivers transport large amounts of vascular plant organic matter into the oceans, resulting in a carbon flux that is estimated to make up approximately 0.7% of terrestrial primary production [7]. The marine environment harbors a large and diverse microbial resource, and some microbial genomes have powerful catabolic pathways to degrade hard-to-biodegrade organic compounds [8,9,10]. It has been reported that many marine fungi play important roles in lignocellulose degradation. For example, ascomycetous fungi *Halosarpheia ratnagiriensis* (strain NIOCC #321) and *Sordaria finicola* (NIOCC #298) mineralized about 9–10% of the U-ring ^14^C-labeled lignin to ^14^CO_2_ within 21 days [11]. However, only a few marine lignin-degrading bacteria were isolated, including species of Actinobacteria and Bacillota phyla and α- and γ-Proteobacteria, such as *Halomonas*, *Arthrobacter*, *Pseudoalteromonas*, *Marinomonas*, *Thalassospira*, *Bacillus*, *Yangia*, *Pelagibaca*, *Salipiger*, *Celeribacter* and *Vibrio* [7,12,13,14,15,16,17,18,19,20,21,22,23]. The extent of marine microbial involvement in lignin degradation and their contribution to the oceanic carbon cycle remains elusive.

*Alteromonadaceae*, a family belonging to γ-Proteobacteria with the type genus *Alteromonas*, was proposed in 2001, which initially included the genera of *Alteromonas*, *Pseudoalteromonas*, *Idiomarina* and *Colwellia* [24]. To date, 25 genera have been proposed belonging to this family in the NCBI taxonomy database, including the genus *Aliiglaciecola*. *Aliiglaciecola* was firstly proposed in 2013 to more appropriately accommodate two recognized species of the genera *Glaciecola* and *Aestuariibacter* (reclassified as *Aliiglaciecola* lipolytica E3^T^ and *Aliiglaciecola* litoralis KMM 3894^T^) [25,26]. Subsequently, *A. coringensis* and *A. aliphaticivorans* were isolated from mangrove forest and sea-tidal flat, respectively [27,28]. Due to limited research, our understanding about the genomic characteristics, metabolic potential and ecological functions of bacteria belonging to the genus *Aliiglaciecola* is still obscure.

In this study, we report a novel strain, *Aliiglaciecola* sp. LCG003, isolated from the intertidal seawater of Lu Chao Harbor, East China Sea, as well as its genome sequences, as the first complete genome of genus *Aliiglaciecola*. Reconstruction of the metabolic pathways, comparative genomics-based approaches and biological experiments were utilized to comprehensively explore the metabolic potentials of strain LCG003, especially in the degradation of lignin, cellulose, protein and other biological macromolecules. The isolation and metabolic characteristics of strain LCG003 present an interesting prospect for the degradation of lignocellulosic material organics, one of Earth’s most abundant biopolymers in the ocean.

## 2. Materials and Methods

### 2.1. Sample Description and Bacterial Isolation

Surface seawater samples were collected in the intertidal zone of Lu Chao Harbor of East China Sea (30.846° N, 121.840° E) in December 2022. The samples were filtered with a pore size of 0.2 μm. The microbes on filter membranes were eluted, centrifuged for 3 min (5000 rpm/min) and resuspended using artificial seawater (ASW) three times. Then, the resuspension was cultured in ASW-lignin (Aladdin, Riverside, CA, USA, CAS 9005-53-2) medium (5 mM lignin, pH = 7.6) at room temperature for three weeks. The enrichments were diluted at a final ratio of 1:10,000 using ASW and cultured on marine 2216E-agar plates (1000 mL seawater, 5 g Peptone, 1 g Yeast Extract and 15 g Agar, pH 7.6–7.7) at 28 °C for two weeks. Finally, the colonies were picked and purified via streaking inoculation. The strain LCG003 (=MCCC 1K08888) was routinely cultivated on 2216E medium under aerobic conditions and stored at −80 °C in liquid medium supplemented with 20% (*v*/*v*) glycerol.

The artificial seawater (ASW) used in this study contained 52 g/L of NaCl, 10 g/L of MgCl_2_·6H_2_O, 8 g/L of Na_2_SO_4_, 2.8 g/L of CaCl_2_·2H_2_O, 1 g/L of KCl, 0.6 g/L of NH_4_Cl, 0.2 g/L of KH_2_PO_4_ and 2 mM of NaHCO_3_. The ASW was autoclaved and supplemented with 1:1000 trace elements, including 30 mg/L of FeCl_3_·6H_2_O, 2 mg/L of MnCl_2_·4H_2_O, 0.23 mg/L of ZnSO_4_·7H_2_O, 0.2 mg/L of CoCl_2_·6H_2_O, 0.1 mg/L of Na_2_MoO_4_·2H_2_O, 0.2 mg/L of Na_2_SeO_3_ and 0.2 mg/L of NiCl_2_·6H_2_O. The supplementary vitamin mixture included 1.8 mg/L of thiamine, 0.0984 mg/L of nicotinic acid, 0.03 mg/L of pantothenic acid, 0.1 mg/L of pyridoxine, 0.0001 mg/L of biotin, 0.001 mg/L of folate, 0.001 mg/L of cobalamin, 0.001 mg/L of myo-inositol and 0.08 mg/L of 4-aminobenzoic acid (final concentration). Trace elements and vitamin mixture were filtered through 0.1 μm filters before usage.

### 2.2. Microbial Utilization of Carbohydrates and Peptones

The microbial utilization of carbohydrates and proteins was tested in triplicate on ASW-agar plates supplemented with peptone, fructose, maltose, glucose, rhamnose, stachyose, cellulose or sucrose as the sole carbon source. The final concentration of peptone and cellulose was 0.5 g/L, and the final concentration of the other carbohydrates was 5 mM. The exponential phase cultures of strain LCG003 cultivated on 2216E (OD_600_ = 0.4) were centrifugally washed three times with ASW in equal volume, and then, 50 μL of the bacterial solution was coated on different plates with a sole carbon source in triplicate and incubated at 28 °C. The plates were photographed after seven days.

### 2.3. Microbial Degradation of Lignin

The microbial degradation of lignin was tested in triplicate in 100 mL of ASW liquid medium supplemented with lignin as the sole carbon source at a final concentration of 5 mM. The exponential phase cultures of strain LCG003 were centrifugally washed with ASW liquid medium three times and then inoculated into the ASW liquid medium with and without lignin, respectively, at 28 °C. Live bacteria were counted using a SYTO 9/PI live/dead bacteria double strain kit at 0 and 7 days, respectively. The degradation of lignin by strain LCG003 was verified by comparing the colony numbers of strain LCG003 in the control and experimental groups. Additionally, the experiment of aniline blue decolorization was performed via inoculating strain LCG003 on 2216E-agar plates with 3% aniline blue added at 28 °C. After 2 days, the decolorized circles around the colonies in the plate medium was observed.

### 2.4. Genomic DNA Extraction, Sequencing and Assembly

The genomic DNA of strain LCG003 was extracted following the method described by Fang et al. [29]. The sequencing of strain LCG003’s genome was conducted by MajorBio (Shanghai Majorbio Bio-pharm Technology Co., Ltd., Shanghai, China) using a combination of the PacBio RS II Single Molecule Real-Time (SMRT) (Pacific Biosciences, Menlo Park, CA, USA) and Illumina HiSeq 2500 platforms (Illumina Inc., San Diego, CA, USA).

For Illumina sequencing, approximately 1 μg of genomic DNA was sheared into 400–500 bp fragments using a Covaris M220 Focused Acoustic Shearer (Covaris, Woburn, MA, USA), according to the manufacturer’s protocol. The sheared fragments were then used to prepare the Illumina sequencing libraries with the NEXTFLEX Rapid DNA-Seq kit (NEXTFLEX, San Jose, CA, USA).

Regarding Pacific Biosciences sequencing, 15 μg of DNA was aliquoted and spun in a Covaris g-TUBE at 6000 RPM for 60 s using an Eppendorf 5424 centrifuge (Eppendorf, Hamburg, Germany). The DNA fragments were subsequently purified, end-repaired and ligated with SMRTbell sequencing adapters following the recommendations of Pacific Biosciences. The resulting sequencing library underwent three rounds of purification using 0.45× volume of Agencourt AMPure XP beads (Agencourt, Scottsdale, AZ, USA), as recommended by the manufacturer. Next, a ~10 kb insert library was prepared and sequenced on one SMRT cell using the standard methods.

The data obtained from both the PacBio and Illumina platforms were utilized for bioinformatics analysis. The complete genome sequence was assembled by combining the reads from PacBio and Illumina. The original image data were converted into sequence data through base calling, resulting in the generation of raw data or raw reads, which were saved as a FASTQ file. Quality trimming was performed to remove low-quality data, and a quality information statistic was employed. The reads were then assembled into a contig using the hierarchical genome assembly process (HGAP) and canu [30]. The final circularization step was manually checked and completed, resulting in a complete genome with a single seamless chromosome. Finally, error correction of the PacBio assembly results was conducted using the Illumina reads with Pilon [31]. The complete genome sequence of *Aliiglaciecola* sp. LCG003 was deposited in the GenBank database under accession number CP128185.1.

### 2.5. Gene Annotation and Genomic Comparison

For ORF prediction and gene annotation, we used the NCBI prokaryotic genome annotation pipeline [32]. The predicted protein sequences were aligned with the Clusters of Orthologous Groups of proteins (COG) [33] and TransporterDB 2.0 [34] databases using BLASTp software [35] (version 2.9.0) (identity, 50%; query-cover, 80%; e-value, 1 × 10^−5^ and score, 40). BlastKOALA [36] was used to assign the Kyoto Encyclopedia of Genes and Genomes (KEGG) annotation. We utilized IslandViewer 4 [37] for the prediction of genomic islands.

To cluster the protein families of strain LCG003 and phylogenetically related strains, we employed a local OrthoMCL 2.0.9 [38]. The clustering was performed using the following cutoff values: identity, 50%; query coverage, 50%; E value, 1 × 10^−10^; score, 40 and MCL inflation, 1.5. Protein families that were only present in one strain were considered strain-specific. To calculate the average nucleotide identity (ANI) and average amino acid identity (AAI), we used fastANI [39] and CompareM (https://github.com/dparks1134/CompareM, accessed on 10 December 2023), respectively, with the default parameters. The genome-to-genome distance (DDH) was calculated using GGDC 3.0 [40].

### 2.6. Phylogenetic Analysis

To study the phylogeny of strain LCG003 and its related strains, we employed 120 conserved bacterial marker genes from the GTDB taxonomy. Firstly, we predicted the sequences of these 120 concentrated proteins in the genomes using GTDB-Tk (database version: Release 07-RS207) [41]. Next, we aligned the sequences individually using Clustal Omega [42]. After manually degapping the aligned sequences, we accounted for any missing concentrated proteins in certain genomes by adding the corresponding number of “-” based on the sequence length after degapping. Subsequently, we tandemly connected the degapped alignments of each concentrated protein [43]. The phylogenetic tree was constructed using the neighbor-joining method with FastTree2 [44]. To evaluate the tree’s robustness, a bootstrap analysis was performed with 1000 replicates. Finally, we plotted the phylogenetic tree using iTOL [45].

## 3. Results and Discussions

### 3.1. Description of Strain LCG003

The samples of intertidal surface seawater from Lu Chao Harbor were incubated with lignin as the sole carbon source at room temperature for two weeks. The enriched microbes were then cultured on marine 2216E-agar plates. A colony, strain LCG003, was picked from a marine 2216E-agar plate, which formed a transparent circular colony with regular and slightly raised edges. Transmission electron micrograph (TEM) showed that the cell length of strain LCG003 was 0.9–1.5 μm, the width was 0.4–0.7 μm and the flagellar length was 1.5–3.5 μm (Appendix A). Cultured in the marine 2216E medium, its optimal growth temperature range was 25–28 °C (Appendix A), and biofilms were observed on the walls of conical flasks after 3 days of cultivation at each temperature.

### 3.2. The Phylogeny of Strain LCG003

The alignments of their 16S rRNA gene sequences showed that strain LCG003 is closely related to *Aliiglaciecola* sp. W161 (OP344131.1), *Aliiglaciecola litoralis* SJS3-28 (MG780337.1), *Aliiglaciecola* sp. M165 (NZ_VKKH01000004.1), *Aliiglaciecola lipolytica* E3^T^ (NR_044430.1) and *Aliiglaciecola lipolytica* D2R05 (NZ_JAHKPT010000031.1), with identities of 99.55%, 97.00%, 96.61%, 96.46% and 96.35%, respectively. Moreover, we acquired the complete genome of strain LCG003 (see below) and constructed a phylogenetic tree based on 120 conserved protein sequences (known as GTDB taxonomy). It showed that strain LCG003 formed a coherent phylogenetic cluster associated with *Aliiglaciecola lipolytica* E3 [25], *Aliiglaciecola lipolytica* D2R05 and *Aliiglaciecola* sp. M165 [46] (Figure 1). In addition, strain LCG003 shared ANIb values of 70.67–70.91 with the other three *Aliiglaciecola* strains, which were higher than those of two neighboring genera, *Aestuariibacter* and *Paraglaciecola* (68.2–69.11) (Appendix A). The AAI values between strain LCG003 and the other *Aliiglaciecola* strains were 73.2 to 73.82 but were, at most, 66.34 with the *Aestuariibacter* and *Paraglaciecola* species (Appendix A). The DDH values between strain LCG003 with and the other *Aliiglaciecola* strains were merely 14.8 to 14.9 (Appendix A). These pieces of evidence indicated that strain LCG003 represents a distinct phyletic lineage in the genus *Aliiglaciecola* belonging to the family *Alteromonadaceae*.

### 3.3. The Genomic Features of Strain LCG003

The complete genome of strain LCG003 consists of one chromosome with a total length of 4,411,932 base pairs (bp), which is smaller than those of closely related strains, e.g., *A. lipolytica* E3, *A. lipolytica* D2R05 and *Aliiglaciecola* sp. M165. The G + C content of strain LCG003 genome is 42.5%. The genome of strain LCG003 contains 3871 genes, including 3771 protein-coding genes, 58 tRNAs, 4 rRNA operons (including 2 copies of 16S-tRNA^Ile^-tRNA^Ala^-23S-5S and 2 copies of 16S-tRNA^Ala^-tRNA^Ile^-23S-5S), 4 ncRNAs and 26 pseudogenes (Table 1).

The graphical representation of the genome is shown in Figure 2. Upon COG classification, 2421 (63.76%) protein-coding genes were classified into 22 categories (Appendix A). The major categories of COG were translation, ribosomal structure and biogenesis (8.20%); general function prediction only (8.09%); amino acid transport and metabolism (7.06%); signal transduction mechanisms (6.55%) and energy production and conversion (6.48%). Moreover, up to 12 genomic islands were found in strain LCG003 (Appendix A). In addition, 60 transposases and 13 integrases and/or recombinases were predicted in strain LCG003, of which 58 (79.45%) were absent in the other three *Aliiglaciecola* strains (Appendix A). These pieces of evidence suggest that horizontal gene transfer (HGT) is important for the speciation of this strain.

### 3.4. The Metabolic Characteristics of Strain LCG003

To study the metabolic characteristics and ecological potentials of strain LCG003, we reconstructed the metabolic pathways and compared them with those of the other three *Aliiglaciecola* strains (Figure 3).

#### 3.4.1. The Utilization of Carbohydrates and Carboxylic Acids

As many as 15 extracellular (signal-fused) and 4 intracellular glycoside hydrolase genes were predicted in strain LCG003. Among these extracellular glycoside hydrolases, 10 genes (66.7%) were absent in the other three *Aliiglaciecola* strains (BLASTp identity, 50%; query coverage, 50% and e-value, 1 × 10^−5^). Also, 11 genes of carbohydrate transporters were predicted in strain LCG003, including glucose, L-rhamnose, maltose/moltooligosaccharide, glycoside/pentoside/hexuronide, arabinose, fucose and glycerol-3-phosphate transporters (Appendix A). Moreover, the complete metabolic pathways of glucose, fructose, sucrose, rhamnose, maltose, stachyose, xylose, starch and cellulose were predicted in strain LCG003. In particular, the genes of L-rhamnose/H+ symporter RhaT (QR722_01220), L-rhamnose isomerase (QR722_01230) and bifunctional rhamnulose-1-phosphate aldolase (QR722_01235) were found only in strain LCG003 but were absent in the other three *Aliiglaciecola* strains, suggesting its unique ecological function of L-rhamnose degradation. Our experiments further confirmed that strain LCG003 could grow with glucose, fructose, sucrose, rhamnose, maltose, stachyose and cellulose used as the sole carbon sources (Figure 4).

Besides carbohydrates, strain LCG003 also contains the transport genes of carboxylic acids, e.g., lactate permease (QR722_RS12480), cation/acetate symporter (QR722_RS15990) and dicarboxylate/amino acid: cation symporter (QR722_RS09570 and QR722_RS10460). Our culture experiments confirmed that strain LCG003 could grow with sodium acetate as the sole carbon source (Figure 4). In addition, polyhydroxybutyrate (PHB) is a microbial bioprocessed polyester belonging to the polyhydroxyalkanoate (PHA) family. The genes of PHB depolymerase (QR722_RS11850) and 3-hydroxybutyrate dehydrogenase (QR722_RS11855) were identified in the four *Aliiglaciecola* genomes, indicating that PHB could also be a carbon source for *Aliiglaciecola* species. These lines of evidence suggested significant ecological functions of strain LCG003 in utilizing carbohydrate and (poly)carboxylic acid degradation.

#### 3.4.2. Metabolism of Amino Acids and Extracellular Proteins

In the aspect of biosynthesis, all four *Aliiglaciecola* strains are able to biosynthesize 20 kinds of amino acids. Also, we predicted seven and six genes involved in the transport of amino acids and oligopeptides in strain LCG003, respectively. Surprisingly, as many as 44 genes of extracellular (SignalP-fused) peptidases (1.16% of all coding genes) were predicted in strain LCG003, which could be classified into 20 peptidase families (Appendix A). The genes of the S9, S8 and M28 family peptidase are the most abundant extracellular peptidases, which include six, five and four copy numbers in strain LCG003.

Peptidase family S9 contains a varied set of serine-dependent peptidases. Most members of the family show restricted specificities and are active mainly on oligopeptides [47,48]. Peptidases in family M28 have co-catalytic zinc ions, which have been confirmed to release a variety of N-terminal amino acids [49]. These peptidase families could be important for strain LCG003 to utilize soluble oligopeptides to satisfy its demand of amino acids.

Furthermore, strain LCG003 appears to have the ability to utilize extracellular insoluble proteins as carbon/nitrogen sources. Peptidase family S8 is the second-largest family of serine peptidases, both in terms of the number of sequences and characterized peptidases. Members of family S8 have a catalytic triad in the order Asp, His and Ser in the sequence, and the majority of them are secreted and nonspecific peptidases that may be associated with nutrient intake. Among these S8 family members, we predicted a polycystic kidney disease (PKD) domain from the gene of QR722_RS06975, which was highly conserved in all *Aliiglaciecola* genomes. The PKD domain as a collagen-binding domain (CBD) can improve the collagenolytic efficiency of the catalytic domain by swelling insoluble collagen and exposing the monomers, as reported in deseasin MCP-01, a bacterial collagenolytic serine protease [50], suggesting the capability of the insoluble collagen degradation of *Aliiglaciecola* strains. Also, we predicted four extracellular metalloproteases with collagenase activity (G3DSA:3.40.390.10), which belong to the M1A/M12B, M3 and M13 families (Appendix A). Such myriad extracellular peptidases illustrated that proteolysis could be one important lifestyle of *Aliiglaciecola* strains.

#### 3.4.3. Strain-Specific Degradation of Lignin and Aromatic Compounds

To further confirm the lignin-degrading capability of strain LCG003, we cultured strain LCG003 in ASW-lignin medium, then counted the live bacteria using a SYTO 9/Pl live/dead bacteria double strain kit. An obvious cell proliferation was observed, indicating that strain LCG003 was capable of lignin degradation (Appendix A). Furthermore, the initiation of lignin degradation depends on the common action of a series of enzymes, including lignin peroxidase (LiP), manganese peroxidase (MnP) and laccase (Lac), which induce the decolorization of aniline blue [51,52]. We observed a strong decolorization by incubating strain LCG003 on aniline blue-containing 2216E-agar plates (Figure 5), which further indicated that strain LCG003 could produce lignin-degrading enzymes. Regarding the genome, a gene of dyp-type peroxidase (encoded by QR722_RS02795) was predicted in strain LCG003, which initiates lignin decomposition via cleavage of the Cα-Cβ linkage of phenolic lignin compounds [53,54,55,56].

In addition, various genes in the degradation of aromatic compounds were predicted, including 4-hydroxybenzoate and vanillate (Appendix A), which were the products of microbial lignin depolymerization [23]. Compared to the other three *Aliiglaciecola* strains, the genome of strain LCG003 contains several unique genes involved in 4-hydroxybenzoate oxidation, including 4-hydroxybenzoate 3-monooxygenase (*pobA*), protocatechuate 4,5-dioxygenase alpha chain (*ligA*) and beta chain (*ligB*), 2-hydroxy-4-carboxymuconate semialdehyde hemiacetal dehydrogenase *(ligC*), 2-pyrone-4,6-dicarboxylate lactonase (*ligI*), 4-oxalomesaconate tautomerase (*galD*), 4-oxalmesaconate hydratase (*ligJ*) and 4-hydroxy-4-methyl-2-oxoglutarate aldolase (*ligK*). Moreover, the genes of the vanillate (4-hydroxy-3-methoxybenzoate) monooxygenase subunits (*vanA* and *vanB*), as well as a GntR family transcriptional regulator in vanillate catabolism, were predicted to be approximately 16 k downstream the 4-hydroxybenzoate-degrading genes (Appendix A). In addition, a strain-specific gene of benzaldehyde dehydrogenase (*xylC*) was predicted, which could transform 4-hydroxybenzaldehyde into 4-hydroxybenzoate. These results suggested that strain LCG003 became a novel lignin and aromatic fragments degrader via horizontal gene transfer and could play an important role in the global cycling of lignocellulosic materials.

#### 3.4.4. The Acquirement of Nitrogen and Phosphorus Sources

For inorganic nitrogen metabolism, we predicted ammonia transporters, nitrate/nitrite ABC transporter, nitrate reductase (encoded by *nasAB*) and nitrite reductase (encoded by *nirBD*) in strain LCG003. Moreover, many organic nitrogen sources, such as nucleoside (QR722_RS15920) and nicotinamide riboside (QR722_RS13445 and QR722_RS15260), could also be imported by strain LCG003. In addition, the extracellular endonuclease gene (QR722_RS15905) and the complete degrading pathways of purines were found in the genome of strain LCG003, indicating that it could obtain nitrogen from biological macromolecules, such as nucleic acids.

For phosphorus utilization, strain LCG003 predictively has the genes of ABC transporters of phosphate, phosphonate and phospholipids. Moreover, it also contains the genes of extracellular alkaline phosphatase (QR722_RS00805, QR722_RS06260 and QR722_RS14025) and phospholipase A (QR722_RS17455). These enzymes may contribute to the survival of strain LCG003 in a phosphorus-limited surface marine environment.

#### 3.4.5. The Biosynthetic Deficiency of Thiamine and Cobalamin

Although we observed the growth of strain LCG003 with many kinds of carbohydrates as the sole carbon sources, our previous studies showed that all tested carbohydrates did not induce its growth without a supplement of mixed vitamins into the culture medium, implying the biosynthetic deficiency of some vitamins in strain LCG003. Indeed, our genomic study revealed that the biosynthetic pathways of thiamine (vitamin B1) were largely incomplete in *Aliiglaciecola* strains LCG003 and M165 hosting with red alga (NCBI BioSample: SAMN12241901) but were complete in strains E3 isolated from coastal surface seawater [57] and D2R05 isolated from an intertidal zone (NCBI BioSample: SAMN19350943). Also, the biosynthetic pathways of thiamine triphosphate (TTP) were also missing in strains LCG003 and M165. TTP is an important coenzyme involved in various biochemical processes in microorganisms, especially in glycolysis and the TCA cycle, e.g., enolase, transketolase, pyruvate decarboxylase, malate dehydrogenase and alpha-ketoglutarate dehydrogenase employ TTP in transferring chemical energy. In this study, no TTP transport gene was predicted in strains LCG003 and M165, suggesting some nonclassical TTP transporters that will be revealed in the future.

Vitamin B12 is a member of the corrinoids that contains a corrin ring, which serves as a cofactor for various enzymes involved in important metabolic reactions, including DNA synthesis, amino acid metabolism and fatty acid synthesis [58]. We found that the biosynthetic pathways of the Vitamin B12 coenzyme (cobalamin) were also missing in the four *Aliiglaciecola* strains. Nevertheless, a gene of the cobalamin-binding protein (QR722_RS17005) was predicted in strain LCG003, which formed an operon with the genes of the iron complex transport system ATP-binding protein (QR722_RS16995) and iron complex permease protein (QR722_RS17000). Taken together, the absence of these coenzyme synthesis capabilities suggested that *Aliiglaciecola* strains have close interactions with other microorganisms in the marine environment.

## 4. Conclusions

In this study, a novel lignin-degrading bacteria strain, LCG003 was isolated from intertidal seawater in Lu Chao Harbor, East China Sea. Phylogenetically, strain LCG003 was affiliated with the genus *Aliiglaciecola* within the family *Alteromonadaceae*. The strain LCG003 genome represents the first complete genome of the genus. Metabolically, strain LCG003 contains many extracellular (signal-fused) glycoside hydrolase genes and carbohydrate transporter genes and can utilize various carbohydrate substrates, including cellulose. Also, strain LCG003 preoccupies various genes of extracellular peptidases, as well as amino acid and oligopeptide transporters, and can grow with peptone as the sole carbon and nitrogen source, indicating a proteolytic lifestyle of *Aliiglaciecola* species. Importantly, strain LCG003 contains a gene of dyp-type peroxidase and strain-specific genes involved in the degradation of 4-hydroxybenzoate and vanillate. Experimentally, the LCG003 strain was confirmed to have a strong decolorization ability for aniline blue and could grow with lignin as the sole carbon source, suggesting an important role in the global carbon cycle of lignocellulosic materials. This study expands our knowledge of lignin-degrading microbial species in marine ecosystems and highlights the contribution of horizontal gene transfer to the degradation of lignocellulosic materials.

## Figures and Tables

**Figure 1 microorganisms-12-00144-f001:**
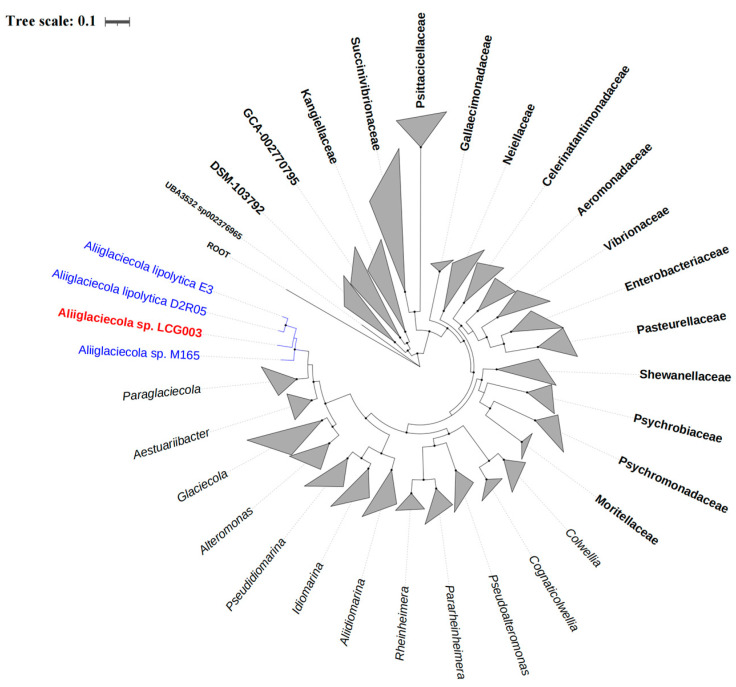
Phylogeny of the *Aliiglaciecola* species and their related species based on 120 concentrated proteins. The *Aliiglaciecola* sp. LCG003 and the other three *Aliiglaciecola* strains are highlighted in red and blue, respectively. Bold characters represent the family under the order Enterobacterales_A, and italic characters represent other genus in the family *Alteromonadaceae*. Black points mean bootstrap values ≥ 60.

**Figure 2 microorganisms-12-00144-f002:**
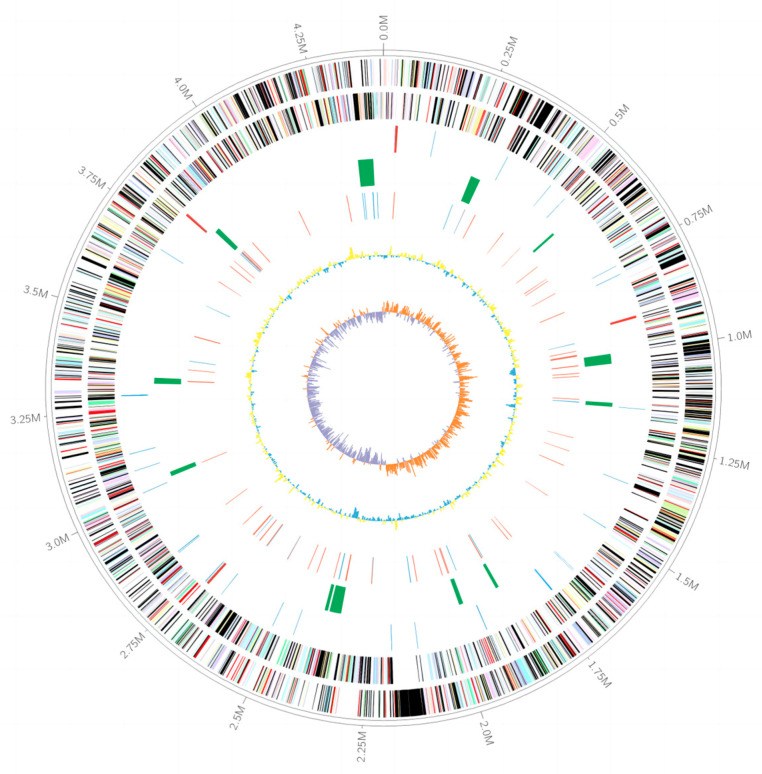
Graphical representation of the *Aliiglaciecola* strain LCG003 genomes. Genes on the forward (shown in outer circle) and reverse (shown in inner circle) strands are colored according to their cluster of homologous genes (COG) categories: RNA genes are highlighted in different colors (tRNAs in blue and rRNAs in red), gene islands are shown in green, genes involved in HGT are shown in different colors (transposase red and integrase/recombinase blue), the GC content is shown in yellow/blue and the GC skew is shown in orange/purple.

**Figure 3 microorganisms-12-00144-f003:**
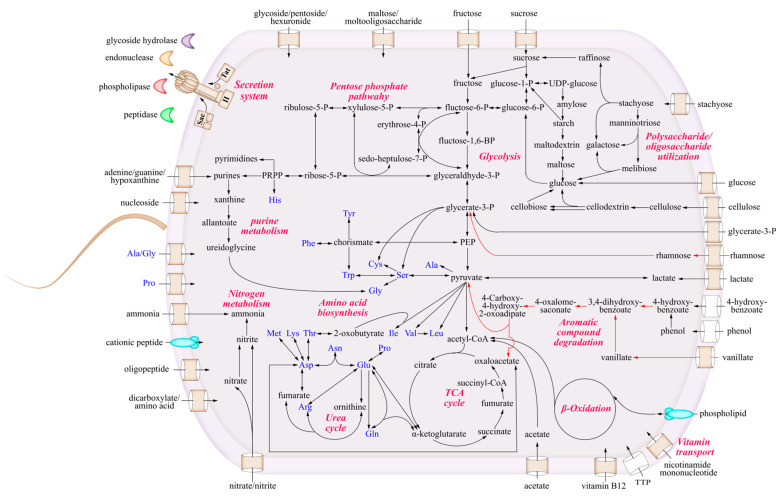
Metabolic potentials of strain LCG003. The black arrows represent metabolic pathways predicted in strain LCG003 and the red arrows highlight the pathways in strain LCG003 that are absent in the other three *Aliiglaciecola* strains.

**Figure 4 microorganisms-12-00144-f004:**
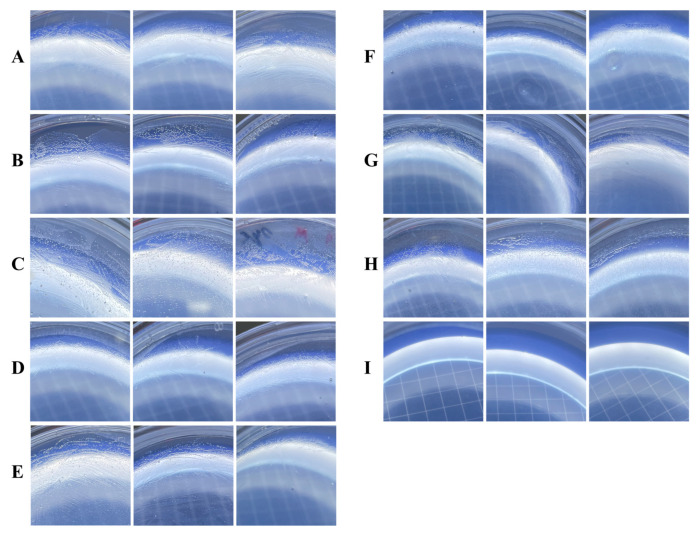
The carbon source utilization of strain LCG003. (**A**–**H**) represents ASM agar plates with glucose, fructose, sucrose, rhamnose, maltose, stachyose, cellulose and sodium acetate as the sole carbon source, respectively, and (**I**) represents the blank control. The plates were photographed after seven days. Each sample was with three independent biological replicates.

**Figure 5 microorganisms-12-00144-f005:**
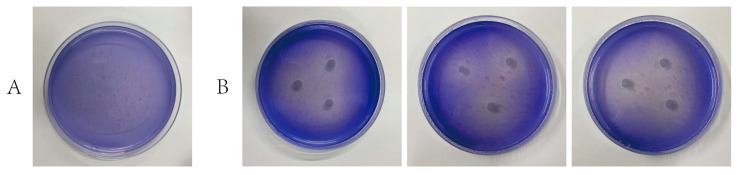
Decolorization of aniline blue by strain LCG003. (**A**) means the blank control, and (**B**) means the decolorization of strain LCG003. Each sample was tested with three independent biological replicates.

**Table 1 microorganisms-12-00144-t001:** Genome features of strain LCG003.

Items	Description
Size (bp)	4,411,932
G + C content (%)	42.5
Coding sequence (%)	87.51
Total genes	3871
Protein-coding genes	3771
Genes assigned to COG	2732
rRNA operons	4
tRNA genes	58
ncRNA genes	4
Pseudogene	26
Gene islands	12

## Data Availability

The GenBank accession number for the complete genome sequence of strain LCG003 is CP128185.1.

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
