# Peer review of "The Phylogeny and Metabolic Potentials of a Lignocellulosic Material-Degrading Aliiglaciecola Bacterium Isolated from Intertidal Seawater in East China Sea"

_microorganisms, 2024, doi:10.3390/microorganisms12010144_

Round 1

Reviewer 1 Report

Comments and Suggestions for Authors

Review

The article by Hongcai Zhang and co-authors “The phylogeny and metabolic potentials of a lignocellulosic material-degrading Aliiglaciecola bacterium isolated from intertidal seawater in East China Sea.” presents the results of isolation, whole-genome sequencing and bioinformatics analysis of the new Aliiglaciecola strain. This strain was isolated from seawater and demonstrated its ability to degrade lignocellulosic material. The presented manuscript fully corresponds to the profile of the journal Microorganisms and is written in good language. The authors conducted significant bioinformatic analysis, which confirms their assumption of the significant environment role of this strain.

I only had a few comments regarding the description of materials and methods.

 Line 78. Please indicate the duration of centrifugation.

 Line 86. The composition of artificial sea water contains agar. However, in a number of experiments the authors use liquid ASW. It would probably be more correct to exclude agar from the ASW composition and list it separately.

Line 88. Please provide a link to the composition of the mixtures of vitamins and microelements.

Line 95. Please indicate what carbon course was used in the liquid medium when culture was grown before seeding on an agar medium with various carbon sources.

Line 95-96. It is written that cells grown to exponential phase were collected by centrifugation and washed. Were the cells resuspended in any volume of medium before being distributed into petri dishes containing agar medium with various carbon sources? If “YES”, please, indicate the OD or g of wet biomass per mL. If “NO”, how did you get an equal number of cells onto the dishes?

Line 219.  The ending of the sentence is missing, a reference to Figure 2 may be needed.

Lines 341-349, Section 3.4.5 requires adjustment. This article is devoted to the characteristics of one strain, LCG003. And the results related to this strain LCG003 are presented. However, new strains, M165, E3 and D2R05, are mentioned in this section. The authors do not provide a link to these strains. Where are they isolated from, what are these strains? If these strains were identified earlier, please provide a link to the corresponding publication.

References. All Latin titles should be italicized. For example, links number 8,14,1516,18, etc.

Author Response

Thank you very much for your questions and suggestions. We are very sorry for the partial revision of your response due to some additional modifications we have made to the content of the article. In response to your suggestions and questions, we have made the following modifications and the modifications are highlighted in red in the article:

Line 78. We have modified in the article “centrifuged for 3 minutes (5,000 rpm/min)”. Please see lines 77.

Line 86. We think your suggestion is very good, and we have deleted “agar”. Please see lines 86.

Line 88. We have supplemented the composition of the vitamins and microelements in the “Materials and Methods” section. Please see lines 88-95.

Line 95. Before seeding on an agar medium with various carbon sources, we used 2216E medium for incubation, which has been added in the article. Please see line 101.

Line 95-96. YES, before dispersing the bacterial solution into petri dishes containing agar medium with various carbon sources, we washed exponential phase cultures of LCG003 strain cultivated on 2216E (OD600 = 0.4) three times by centrifugation with an equal volume of ASW, and then 50 μL of the bacterial solution was coated on different plates of sole carbon source. Please see lines 100-104.

Line 219. Figures 2 in this article was drawn by ourselves, so we did not provide references. And we moved that sentence to the beginning of the paragraph. We also uploaded all the high-resolution Figures to a zip file called “Figures and Tables”. Please see lines 217.

Lines 341-349. Strains M165, E3 and D2R05 have all been isolated by others, with strain M165 hosting red alga, strain E3 isolated from coastal surface seawater and strain D2R05 from the intertidal zone. Their complete genomic data have been uploaded to the NCBI database, and we provide NCBI BioSample numbers and references for these bacteria in the article. See lines 191-192 and 354-356.

References. We have modified all Latin titles into italicized formatting. We have marked the modifications in red.

Reviewer 2 Report

Comments and Suggestions for Authors

I have gone through the manuscript entitled “The phylogeny and metabolic potentials of a lignocellulosic material-degrading Aliiglaciecola bacterium isolated from intertidal seawater in East China Sea”. In this study, authors have isolated a novel strain Aliiglaciecola sp. LCG003 (and its whole genome), isolated from intertidal seawater of Lu Chao Harbor, East China Sea. Additionally, metabolic potentials of this strain, especially in degradation of lignin, cellulose, protein, and other biological macromolecules, are also reported. The manuscript is well design and properly written. However, some aspects must be clarified for a better understanding of the topic and, therefore, I suggest accepting this paper in Microorganisms after a major revision.  

1. Many errors, related to not founding the source reference, have been identified throughout the text. These errors need to be rectified by providing the accurate references.

2. The quality of the figures mut be improved. Moreover, Additionally, it is essential to ascertain whether the authors personally created figures 1, 2, and 3. The figures included in the manuscript should either be self-created or accompanied by the correct references.

3. Captions for supplementary figures and tables are missing.

4. Authors claim that metabolic potentials of this strain have been evaluated, however, a substantial portion of the evaluation relies on predictions. Therefore, in the 2.5 section of the Materials and Methods, additional details about the parameters employed for these predictions must be supplied. Furthermore, confident values corresponding to these predictions should be included.

5. Greater emphasis should be placed on the experimental work conducted to evaluate the metabolism of carbohydrates and carboxylic acids. In this sense I suggest adding the Figure S3 to the manuscript. Additionally, the results related to this figure must be further explained in the manuscript, since they provide stronger evidence than the computational predictions.

Comments on the Quality of English Language

Overall English language and style is good, but some minor errors have been found. So, I recommend careful revision before publishing.

Author Response

Thank you very much for your questions and suggestions. We are very sorry for the partial revision of your response due to some additional modifications we have made to the content of the article. In response to your suggestions and questions, we have made the following modifications and the modifications are highlighted in red in the article:

Regarding questions 1 and 2, Figures 1, 2, and 3 in this article were drawn by ourselves, so we did not provide references. And we uploaded the high-resolution figures to a zip file named “Figures and Tables”.

Regarding question 3, we uploaded the file about the captions for supplementary figures and tables in our supplementary material as "Supplementary Tables and Figures Legends".

Regarding question 4, In Section 2.5, we provide additional information about the parameters used in these predictions. Please see lines 149.

Regarding suggestion 5, we have move Figure S3 to the main text as Figure 4, with a detailed explanation of the results related to the figure. Please see lines 258-261.